# Directed Evolution of Near-Infrared Serotonin Nanosensors with Machine Learning-Based Screening

**DOI:** 10.3390/nano14030247

**Published:** 2024-01-23

**Authors:** Seonghyeon An, Yeongjoo Suh, Payam Kelich, Dakyeon Lee, Lela Vukovic, Sanghwa Jeong

**Affiliations:** 1Department of Biomedical Convergence Engineering, Pusan National University, Yangsan 50612, Republic of Korea; 2Department of Chemistry and Biochemistry, University of Texas at El Paso, El Paso, TX 79968, USA; 3Department of Chemistry, Pohang University of Science and Technology (POSTECH), Pohang 37673, Republic of Korea

**Keywords:** directed evolution, carbon nanotube, serotonin, nanosensor, machine learning, fluorescence, near-infrared

## Abstract

In this study, we employed a novel approach to improve the serotonin-responsive ssDNA-wrapped single-walled carbon nanotube (ssDNA-SWCNT) nanosensors, combining directed evolution and machine learning-based prediction. Our iterative optimization process is aimed at the sensitivity and selectivity of ssDNA-SWCNT nanosensors. In the three rounds for higher serotonin sensitivity, we substantially improved sensitivity, achieving a remarkable 2.5-fold enhancement in fluorescence response compared to the original sequence. Following this, we directed our efforts towards selectivity for serotonin over dopamine in the two rounds. Despite the structural similarity between these neurotransmitters, we achieved a 1.6-fold increase in selectivity. This innovative methodology, offering high-throughput screening of mutated sequences, marks a significant advancement in biosensor development. The top-performing nanosensors, N2-1 (sensitivity) and L1-14 (selectivity) present promising reference sequences for future studies involving serotonin detection.

## 1. Introduction

Optical biosensors are techniques that enable the detection of biological and chemical substances [1] and have many advantages over conventional analytical methods, such as being highly sensitive, specific, and cost-effective [2,3]. Carbon nanotubes, a novel nanomaterial used in these biosensors, are promising building blocks for biosensor applications with unique electronic and optical properties [4]. Among them, single-walled carbon nanotubes (SWCNTs) are 1D nanomaterials with a graphene layer rolled into a tube [5], which has a high surface-to-volume ratio, and is advantageous for surface functionalization [6]. SWCNTs are functionalized with single-stranded DNA (ssDNA), a promising strategy among various functionalizations in biotechnology, and can be dispersed in an aqueous phase and emit fluorescence in the near-infrared region. Especially, the second near-infrared (nIR-II) fluorescence of SWCNT, whose wavelength is around 1000–1700 nm, is useful for detection and imaging as it exhibits very low background and high penetration depths into biological tissues [7]. Most organic fluorophores suffer from photobleaching under high and long illumination. However, even at high fluences, SWCNTs show continuous photostability in fluorescence emission, which is adequate for long-life sensors [8]. The diverse chirality of SWCNTs facilitates high throughput screening and hyperspectral imaging and can be utilized as optical sensors for human disease biomarkers as their different chiral properties in the same dispersion solution allow them to react differently to target analytes [9].

We sought to utilize these SWCNT biosensors to examine their fluorescence response to neurotransmitters, specifically serotonin (5-hydroxytryptamine, 5HT). 5HT is a biomolecule that acts as a neurotransmitter and is known to be associated with depression, mania, and anxiety disorders [10,11,12]. To efficiently sense 5HT, SWCNTs can be functionalized with a specific ssDNA sequence. The ssDNA-wrapped SWCNT complexes formed should ideally recognize only the specific molecules and report their detection optically, via fluorescence changes in response to the target substance [13,14].

To develop the novel 5HT optical nanosensors with high sensitivity and selectivity, we applied the directed evolution protocol to improve the sensor performance of certain ssDNA-SWCNT complexes. Directed evolution is a method mainly utilized in protein engineering that mimics the process of natural selection to manipulate proteins or nucleic acids toward a user-defined goal [15,16,17]. We started with the 5HT-responsive sequence found in our previous report as the original sequence [13], then applied the directed evolution techniques to develop higher-performance ssDNA sequences for 5HT nanosensors [18] (Figure 1). Using a similar method, Lambert et al. also applied directed evolution to find the dopamine (DA)-sensitive SWCNT nanosensor [18]; the team improved the sensor properties of the DNA-wrapped SWCNT through DNA mutations. They used tens of ssDNAs randomly mutated from the original sequence to screen the fluorescence response of the ssDNA-SWCNT complexes upon the addition of DA. Several mutants in each cycle exhibited fluorescence enhancement compared to the initial ssDNA-SWCNT complex and the best mutated sequence was selected.

Following the previous research, we did the modified directed evolution with 3-base mutation for a total of 3 rounds via machine learning (ML) prediction. The total number of 3-base mutated ssDNA sequences is very large, up to 22,032 sequences per round; it is impractical to test all of them. Therefore, we applied an ML model that predicts 5HT-specific sequences from a previous paper to determine the ΔF/F_0_ (increase in fluorescence intensity of 5HT over DI water) results at 1195 nm for each sequence and performed experiments on the top 20 sequences [19]. In this paper, we repeated the evolution process of finding sequences that are responsive to 5HT in terms of sensitivity and selectivity. In this approach, the combination strategy of directed evolution and ML model prediction for fluorescence measurements using ssDNA-SWCNTs offers great potential in terms of time and cost-effectiveness and high throughput in finding the most sensitive sequences for target substances [20].

## 2. Materials and Methods

### 2.1. Materials

Super-purified HiPCo SWCNTs (Batch #HS37-027) were purchased from NanoIntegris (Boisbriand, QC, Canada). DI water, serotonin hydrochloride, dopamine hydrochloride, acetylcholine chloride, γ-aminobutyric acid (GABA), glutamate (Glu), uric acid (UA), and ascorbic acid (AA) were purchased from Sigma-Aldrich(St. Louis, MO, USA). All the DNA molecules were purchased and used without further purification from Integrated DNA Technologies (IDT, Clarville, IA, USA).

### 2.2. Fabrication of ssDNA-SWCNT Nanosensor

The nanosensor dispersion was fabricated as reported previously in the literature [13]. The mixture solution was prepared by adding 1.0 mg ssDNA and 1.0 mg SWCNT to a 2-mL e-tube and filling the remaining volume with 1× PBS to a total volume of 1 mL. The mixture was bath-sonicated for 3 min to ensure that the SWCNTs were well dispersed in the e-tube. The microtube was then tip-sonicated in an ice bath at 50% amplitude for 20 min (VCX-130, SONICS & Materials, Inc., Newtown, CT, USA). After sonication, the microtubes were centrifuged at 21,000× *g* for 1 h, and 800 μL of the supernatant was collected out of a total volume of 1 mL to remove sediment. The final ssDNA-SWCNT suspension was diluted 100 times (10 μL of the sensor to 990 μL of DI water) to measure the absorbance. The concentration of the ssDNA-SWCNT complex was calculated by measuring its absorbance at 632 nm (extinction coefficient = 0.036 mg·L^−1^ cm^−1^) [21].

### 2.3. Fluorescence Imaging of ssDNA-SWCNT Nanosensors Immobilized on a Glass Substrate

A glass coverslip (thickness = 0.13 mm, Ø18) is soaked in 10% APTES in anhydrous ethanol for 10 min. Then the coverslip was rinsed with DI water and blown with N2 gas. The APTES-coated glass coverslip was immersed in 50 mg/L of SWCNT and incubated at 4 °C for 30 min. After incubation, the coverslip was softly rinsed with DI water. The coverslip was then fixed in the imaging chamber (Chamlide-AC, Live Cell Instrument, Seoul, Republic of Korea). The surface-immobilized nanosensors were imaged on an inverted fluorescence microscope with a 50× objective lens by 721 nm laser excitation [13]. The fluorescence image was captured with a NINOX 640 camera (Raptor Photonics, Larne, UK) through a 950-nm long pass filter. Before the 5HT addition, the imaging chamber was filled with 990 μL of 1× PBS. After 30 s, 10 μL of 10 mM 5HT was injected into the imaging chamber for a final 5HT concentration of 100 µM. Fluorescence images were captured at 0.5 s intervals with exposure time = 500 ms.

### 2.4. Fluorescence Measurement

The ssDNA-SWCNT complex suspension was diluted to 10 mg/L by 1× PBS before the fluorescence measurement. 5HT is the main substance used in terms of sensitivity, and DA, acetylcholine (Ach), GABA, Glu, UA, and AA were used to compare the fluorescence intensity to 5HT in terms of selectivity. Analytes were dissolved in DI water to make a stock solution, with concentrations of 1 mM DA, 5HT, Ach, GABA, Glu, AA, and 100 µM UA. Because of the low solubility of UA, 100 μM UA stock solution was used. 50 µL of each stock solution was added to 450 µL of sensor solution for a final analyte concentration of 100 µM except 10 µM for UA. 50 µL of DI water was added to the sensor solution for measuring baseline fluorescence. The fluorescence was measured after 20 min incubation and the change in the fluorescence spectrum was measured. A 721-nm laser (PSU-H-LED laser, CNI laser, Changchun, China) was used as an excitation light source, and detection was performed using an InGaAs photodiode array detector (NIRQuest, Ocean Insight, Orlando, FL, USA). Fluorescence measurements were performed in triplicate, and the average value was calculated and used for data analysis.

### 2.5. Directed Evolution with Machine Learning Screening

To find sensors that are highly sensitive and selective to 5HT, we used a directed evolution approach. First, the initial 30-mer ssDNA sequence 5′-CCCCCCAGCCCTTCACCACCAACTCCCCCCCC-3′, which is known to have a fluorescent response to 5HT [13], was used as the original sequence. Three bases in the center 18 bases were randomly mutated to the one of A, G, C, or T except for two fixed 6-mer (C)_6_ on each flank to obtain a diverse DNA pool. This yielded a total of 22,032 sequences. For ML, we applied multiple ML regression models to obtain the ensemble-predicted ΔF/F_0_ values, as reported in Ref [19]. This ensemble model method predicts ΔF/F_0_ values for DNA-SWCNT complexes with the defined DNA sequences. We then ranked the sequences according to the responsiveness to 5HT and ensured that an ensemble of ML classification models based on convolutional neural networks (developed and validated in Ref [19]) also classified the highest-ranked sequences as high response sequences. The ensemble model produced the multiple predicted ΔF/F_0_ for 5HT, and we calculated the mean value as the indicator (mean_r). The top 20 sequences with the highest predicted ΔF/F_0_ were chosen for experimental characterization [19]. We experimentally measured the change in fluorescence in response to 5HT and DA. Among the 20 sensors, the best-performing sequence was selected and used as the starting sequence for the next round. The sensor performance was compared in terms of sensitivity and selectivity, and each performance parameter was calculated based on the fluorescence peak at 1202 nm. Depending on the performance, we aimed for a total of three rounds.

## 3. Results

### 3.1. Directed Evolution with Machine Learning Prediction

To fabricate a sensor with better performance for 5HT, we applied directed evolution technology for the optimal ssDNA-SWCNT complex sensor. Directed evolution is a technique used in protein engineering that mimics the process of natural selection to guide the evolution of high-affinity proteins or nucleic acids to the target molecules [21]. Based on a previous paper that applied this technology to ssDNA, we concurrently mutate and test the ssDNA sequences with SWCNT to find a sequence that is more sensitive in response to 5HT (Figure 1). Using the original sequence 5′-CCCCCCAGCCCTTCACCACCAACTCCCCCC-3′ [13], which is known to respond to 5HT in the previous paper [19], we mutated 3 bases from the original sequence to produce 22,032 sequences, as the first mutated DNA library (Figure 1a). With this mutated library, we applied the ensemble ML model from our previous work [19] to predict the fluorescence response of each ssDNA-SWCNT complex to 5HT. This ensemble ML model consists of 2 regression models based on support vector machine approach using radial basis function (RBF) and sigmoid kernels and 9 convolutional neural network classification models. This ensemble ML model (regression) produces the min, max, and mean values of predicted fluorescence change of ssDNA-SWCNT complex to 5HT as ΔF/F_0_ (ΔF/F_0_ = (F − F_0_)/F_0_, where F is the fluorescence intensity for the 5HT and F_0_ is the baseline fluorescence intensity before 5HT addition). Among the predicted ΔF/F_0_ and corresponding sequence, the top 20 sequences with the highest ΔF/F_0_ were selected by using the mean value as an indicator (Figure 1b,c). The same sequences were also checked to be high response to 5HT using the classification models. We experimentally fabricated ssDNA-SWCNT complexes with 20 new ssDNA sequences and measured their ΔF/F_0_ to find out the best-performing sequence. This best sequence is used as the starting sequence for the next round of directed evolution.

### 3.2. Towards 5HT Nanosensors with Higher Sensitivity

At first, we applied this directed evolution strategy to find the better 5HT nanosensors in terms of sensitivity. We measured the 5HT responsive fluorescence change of ssDNA-SWCNT complexes from the top 20 sequences of 22,032 mutated sequences. Sensitivity to 5HT in those sensors could be compared by ΔF/F_0_ values at 1202 nm (Figure 2a). (ΔF/F_0_ = (F − F_0_)/F_0_, where F is the fluorescence intensity after 5HT addition, and F_0_ is the fluorescence intensity after DI water addition.) Among the 20 sequences, the N1-12 sequence showed the highest ΔF/F_0_ of 2.366, which is 2.5 times larger than ΔF/F_0_ of 0.949 from the original sequence. The N1-12 sequence was used as the starting sequence for round 2 of ML (Appendix A), and the top 20 of the 22,032 mutated sequences from N1-12 were tested in the same protocol. The best sequence in round 2, N2-1, showed an improved ΔF/F_0_ of 3.705 (Appendix A). Using N2-1 as the starting sequence for round 3, we found another mutated sequence of N3-5 with a slightly improved ΔF/F_0_ of 3.765. ΔF/F_0_ values from all tested sequences were plotted in Figure 2b. Sensitivity improvement by directed evolution seems to be saturated at round 2 because the ΔF/F_0_ of N2-1 and N3-5 do not show a statistically significant difference (Figure 2c). Our directed evolution protocol improves the ΔF/F_0_ of the ssDNA-SWCNT nanosensor from ΔF/F_0_ = 0.949 to ΔF/F_0_ = 3.765, and this evolution is almost saturated at round 2. Experiments were conducted to compare the performance of the sensors made from the best-performing ssDNA in each round. The response of the sensors to 5HT in 1× PBS was compared. The normalized ΔF/F_0_ values for 5HT concentrations of 0.1 μM, 1 μM, 5 μM, 10 μM, 30 μM, 70 μM, and 100 μM are shown experimentally (Figure 2d). *K_d_* values were obtained via the Hill equation [22].

Hill equation:(1)$$∆F/F_{0}=\frac{{\left(∆F/F_{0}\right)}_{max\;}\times {\left[5HT\right]}^{n}}{K_{d}+{\left[5HT\right]}^{n}},$$

*K_d_* = dissociation constant, *n* = cooperativity.

The dissociation constant (*K_d_*) indicates the strength of binding between ssDNA-SWCNT and the analyte of 5HT. The smaller the *K_d_* value, the higher the binding force between the two, and the larger the *K_d_* value, the lower the binding force between the two. According to the hill equation, the *K_d_* values of the original, N1-12, N2-1, and N3-5 are 11.6 μM, 12.9 μM, 10.7 μM, and 6.6 μM, respectively. Those nanosensors showed similar *K_d_* around ~10 μM. Also, the *K_d_* value is decreasing as the number of rounds increases.

### 3.3. Towards 5HT Nanosensors with Higher Selectivity

Herein, ‘selectivity’ refers to the ratio, which represents the fluorescence increase for DA and 5HT calculated as 5HT/DA (DA to 5HT fluorescence increase ratio). A higher value indicates that the nanosensor is more sensitive to 5HT than DA, i.e., having a good performance for 5HT.

Following the sensitivity test, we compared the fluorescence changes with DA to round-find sequences that respond specifically to 5HT. Since 5HT and DA have similar chemical structures, there is cross-sensitivity in fluorescence intensity. To avoid this cross-sensitivity, we wanted to check the performance of the sensor from the selectivity point of view and find a sequence with high selectivity for 5HT. We analyzed the data by measuring the reactivity to 5HT and the reactivity to DA and calculating the ratios. We measured the ΔF_5HT_/F_0_ and ΔF_DA_/F_0_ values at 1202 nm (Figure 3a,b). and divided them to obtain the 5HT/DA value. Selectivity was compared by the fluorescence intensity value of 5HT/DA, which is the ratio of the fluorescence change measured at the 1202 nm wavelength. The selectivity was defined as 5HT/DA = {(F_5HT_ − F_0_)/F_0_}/{(F_DA_ − F_0_)/F_0_} where F_5HT_ is the fluorescence intensity after 5HT addition, F_DA_ is the fluorescence intensity after DA addition, and F_0_ is the baseline fluorescence intensity at 1202 nm. Among the 20 sequences, the L1-14 sequence showed the highest 5HT/DA of 0.799, which is 1.6 times larger than the 5HT/DA of 0.508 from the original sequence. L1-14 sequence was used as the starting sequence for round 2, and the top 20 of the 22,032 mutated sequences from L1-14 were tested in the same protocol (Appendix A). The best sequence in round 2, N2-6, showed an improved 5HT/DA of 0.592. 5HT/DA values from all tested sequences were plotted in Figure 3c. Selectivity improvement by directed evolution seems to be saturated at round 1 because the ΔF_5HT_/F_DA_ of N2-6 shows a statistically significant difference but does not show an increase in performance over the original in round 1 (Figure 3d). We measured the selectivity of our fluorescent nanosensors to different neurochemicals and biomolecules (Figure 3e). When checking the selectivity for original and L1-14, both sensors show minimal response to Ach, GABA, Glu, and UA. AA shows a larger response to the sensor compared to DA, but the effect is lower at L1-14. AA is already known to increase the fluorescence of ssDNA functionalized SWCNT [23], naturally, our experiments showed a higher ΔF/F_0_ for AA than for DA. Our directed evolution protocol improves the ΔF_5HT_/F_DA_ of ssDNA-SWCNT nanosensor from ΔF_5HT_/F_DA_ = 0.508 to ΔF_5HT_/F_DA_ = 0.799, and this evolution is almost saturated at round 1. Compared to the original sequence, the L1-14 sequence has a 3 bp mutation at positions 4, 7, and 11, all of which are A mutations (Figure 3f).

### 3.4. Fluorescence Image Analysis of Surface-Immobilized Nanosensors after 5HT Treatment

Fluorescence images were obtained for the N2-1 ssDNA-SWCNT nanosensor, which showed the best performance in sensitivity. SWCNTs were adsorbed to an APTES-coated glass coverslip, filled with 1× PBS, and spiked with 5HT to observe the fluorescence response. 5HT was injected at 30 s for a final concentration of 100 μM and the change in ΔF/F_0_ was observed for 1 min. Overall, we can observe the fluorescence increase upon treatment with 5HT (Figure 4a,b). On average, the nanosensor fluorescence increased up to 40% ΔF/F_0_ after 5HT treatment (Figure 4c). Since the sensors showed good responsiveness to 5HT while anchored to the surface, it is expected that the sensors will also show good response to neurotransmitters when coated around the neural cells.

## 4. Discussion

In this study, we tested 100 mutated sequences in multiple rounds of evolution (total sensitivity 3 rounds + selectivity 2 rounds = 5 rounds) from the original ssDNA sequence. By overcoming the physical limitations of sequence-by-sequence testing using conventional directed evolution, ML allowed us to scale up our experimental approach. The high throughput 5HT sensors we found are expected to serve as real-time sensing for neurodegenerative disorders including Parkinson’s disease, Huntington’s disease, and schizophrenia.

In the ssDNA functionalized SWCNT sensor, the sensitivity and selectivity have been improved in various methods. Specifically, the introduction of XNA, a chemically modified form of DNA, has resulted in an improved turn-on response and increased stability [24]. However, the chemical modification of DNA, such as XNA preparation, incurs higher costs and longer testing times for each sample. Additionally, Lambert et al. reported the application of directed evolution to the dopamine nanosensor for higher sensitivity in ssDNA-SWCNT [18]. However, it is noteworthy that the conventional approach involved screening the mutated DNA library through random selection.

In contrast, our methodology integrates directed evolution with machine learning screening, enabling a cost-effective and time-efficient exploration of diverse DNA-assisted binding moieties on SWCNT. In comparison to previous directed evolution methods for ssDNA-SWCNT, our approach utilizes a machine learning model to predict more sensitive mutated sequences for 5HT. This enhances our ability to discover superior sequences, offering a more targeted approach compared to the random selection of mutated sequences.

The experimental results showed that there were commonly mutated bases, such as the seventh base of the sequences that were predicted to have high performance, and that the sixth base sequence was mutated in the order of G → T → A during the three rounds of sensitivity. In addition, the mutated bases were mostly changed to adenine, and in the final selected sequences, N2-1 at sensitivity and L1-14 at selectivity, three bases in the mutation site were changed to A. In addition to the original sequence, which was used as a reference in this study, we expect to find a sequence that performs better by conducting similar experiments on other sequences known to bind well to 5HT, such as 5HT aptamer [14,25]. Finally, analyzing these selected sequences will enable the discovery of 5HT sensors based on performance, aiding in more accurate predictions from an ML perspective. The biomedical application of SWCNT faces challenges, including concerns about toxicity, the necessity for improved biocompatibility, and ensuring long-term stability. Addressing issues related to biodistribution, immunogenicity, production scalability, and ethical considerations is crucial for advancing the responsible use of SWCNTs in biomedical engineering. Ongoing research efforts are essential to overcome these limitations and unlock the full potential of SWCNTs in clinical applications.

## 5. Conclusions

In summary, we used a combination of measuring the neurotransmitter fluorescence with ssDNA-SWCNT complexes and directed evolution technology to evaluate the ssDNA-SWCNT sensor performance by comparing the fluorescence response to randomly mutated sequences. The final selected sequences, N2-1 (5′-CCCCCCAACCCTACACAACCACCTCCCCCC-3′) and L1-14 (5′-CCCCCCAGCACTACACAACCAACTCCCCCC-3′) can serve as reference sequences for future studies using ssDNA-SWCNT sensors to detect 5HT. In this study, we did not find any sequences with a selectivity greater than 1.0, meaning that the fluorescence intensity for 5HT exceeded that for DA. In this regard, we believe that conducting the same study for other neurotransmitters (acetylcholine, norepinephrine, GABA, etc.) in terms of selectivity will enhance our understanding of 5HT and allow us to discover sensors with superior performance.

## Data Availability

Data can be shared upon request to Sanghwa Jeong.

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
