# Peer review of "Directed Evolution of Near-Infrared Serotonin Nanosensors with Machine Learning-Based Screening"

_nanomaterials, 2024, doi:10.3390/nano14030247_

Round 1

Reviewer 1 Report (Previous Reviewer 1)

Comments and Suggestions for Authors

The authors have addressed the concerns and the paper has been improved. I think this work can be accepted for publication.

Author Response

Thank you for kind comments.

Reviewer 2 Report (Previous Reviewer 2)

Comments and Suggestions for Authors

Accept.

Author Response

Thank you for kind comments.

Reviewer 3 Report (New Reviewer)

Comments and Suggestions for Authors

The manuscript titled "Directed Evolution of Near-infrared Serotonin Nanosensors with Machine Learning-based Screening" combines directed evolution and machine learning-based prediction to improve the sensitivity and selectivity of ssDNA-SWCNT nanosensors, and the best performing nanosensor sequence were N2-1 (sensitivity) and L1-14 (selectivity). The experimental design is comprehensive and complete with detailed experimental data. In addition, there are some details that need attention and the manuscript needs stronger editing. Therefore, this manuscript is suitable for publication on Nanomaterials.

The comments are listed in detail as follows.

1.       In parts 2.3 and 2.4, 721 nm was chosen as the excitation wavelength, can this wavelength be further optimised subsequently?

2.       Why were DA, Ach, GABA, Glu, UA, and AA chosen for the sensor selectivity experiments? What is the basis for the selection?

3.       What is the lifetime stability of the sensor? Have any relevant experimental studies been conducted?

4.       Is there a difference between Figure 2a and 3b?

5.       It is advised that the first two bars of Figure 3e be supplemented with error bar labelling?

6.       The word L1-14 seems to have been omitted before the second bracket in line 349.

7.       The article did not conduct any exploratory experiments related to the application of actual samples of this sensor, what do you think are the possible limitations of this sensor in its future application to neurodegenerative diseases?

Author Response

For reviewer 3:

The manuscript titled "Directed Evolution of Near-infrared Serotonin Nanosensors with Machine Learning-based Screening" combines directed evolution and machine learning-based prediction to improve the sensitivity and selectivity of ssDNA-SWCNT nanosensors, and the best performing nanosensor sequence were N2-1 (sensitivity) and L1-14 (selectivity). The experimental design is comprehensive and complete with detailed experimental data. In addition, there are some details that need attention and the manuscript needs stronger editing. Therefore, this manuscript is suitable for publication on Nanomaterials.

The comments are listed in detail as follows.

Comment 1: In parts 2.3 and 2.4, 721 nm was chosen as the excitation wavelength, can this wavelength be further optimised subsequently?

Answer: We greatly appreciate your kind words. For colloidal SWCNT suspensions, ~721-nm excitation wavelength has been widely used in previous researches, including our work (Jeong, S.; Yang, D.; Beyene, A.G.; Bonis-O'Donnell, J.T.D.; Gest, A.M.M.; Navarro, N.; Sun, X.; Landry, M.P. High-throughput evolution of near-infrared serotonin nanosensors. Sci. Adv. 2019, 5, eaay3771.). It is possible to use other wavelengths as excitation wavelengths, but in this case, the SWCNTs of different chirality is excited, so their fluorescence spectra will change  (Gillen, A.J.; Antonucci, A.; Reggente, M.; Morales, D.; Boghossian, A.A. Distinguishing dopamine and calcium responses using XNA-nanotube sensors for improved neurochemical sensing. bioRxiv. 2021, 2021.02.20.428669.). We used the 721-nm wavelength for the consistency for our previous researches and other work in our field.

Comment 2. Why were DA, Ach, GABA, Glu, UA, and AA chosen for the sensor selectivity experiments? What is the basis for the selection?

Answer: Thank you for your consideration in this matter. DA, Ach, GABA, and Glu are neurotransmitters, as are 5HT and DA, so we thought they were appropriate for measuring selectivity. UA and AA were measured at the request of another reviewer, and we think they were requested because they are small biomolecules that co-occur in biological fluids (Balamuruganm, J.; Senthil Kumar, S.M.; Thangamuthu, R.; Pandurangan, A. Facile and controlled growth of SWCNT on well-dispersed Ni-SBA-15 for an efficient electro-catalytic oxidation of ascorbic acid, dopamine and uric acid. J Mol Catal A Chem. 2013, 372, 13-22). In addition, all of these substances above are interfering compounds in biological systems for 5HT and DA, so it is appropriate to use these substances to measure selectivity for 5HT for selectivity with 5HT.

Comment 3. What is the lifetime stability of the sensor? Have any relevant experimental studies been conducted?

Answer: Thank you for your attention to this matter. Fluorescent SWCNT colloids have been known to have high photostability than other fluorophores such as organic dyes. For example, surfactant-encapsulated SWCTN suspension showed the negligible photobleaching properties, even though near-infrared fluorescence of ICG dye significantly decreases in the same excitation condition. We have revised the manuscript to include the stability as below with proper reference (Boghossian , A.A.; Zhang , J., Barone; P.W.; Reuel; N.F.; Kim; J.-H.; Heller; D.A.; Ahn; J.-H.; Hilmer; A.J.; Rwei; A.; Arkalgud; J.R.; Zhang; C.T. and Strano; M.S. Near-Infrared Fluorescent Sensors based on Single-Walled Carbon Nanotubes for Life Sciences Applications. ChemSusChem. 2011, 4, 848-863.).

Changes to manuscript:

Introduction

Most of organic fluorophores suffer from photobleaching under high and long illumination. However, even at high fluences, SWCNTs show continuous photostability in fluorescence emission, which is adequate for long-life sensors [8].

Comment 4. Is there a difference between Figure 2a and 3b?

Answer: Thank you for your comment. Figure 2a and 3b are graphs of fluorescence spectra for a SWCNT sensor using the same original sequence. However, there might be batch-to-batch difference between DNA-SWCNT samples. The relative fluorescence change in response to serotonin is almost similar but the absolute intensity of each sensor is different. Therefore, we include the fluorescence spectra of the same DNA-SWCNT sample for the response to serotonin and dopamine in Figure 3a and Figure 3b.

Comment 5. It is advised that the first two bars of Figure 3e be supplemented with error bar labelling?

Answer: We're grateful for your kind offer. We revised the error bar in Figure 4e as below.

Changes to manuscript:

Comment 6. The word “L1-14” seems to have been omitted before the second bracket in line 349.

Answer: Thank you for raising your concerns. As your comment 6, we have noticed that the words "L1-14" are missing before the L1-14 sequence in the conclusion section on line 349. Again, thank you for your detailed comments. We have corrected the omission and you can see the revisions below.

Changes to manuscript:

The final selected sequences, N2-1 (5'-CCCCCCAACCCTACACAACCACCTCCCCCC-3') and L1-14 (5'-CCCCCCAGCACTACACAACCAACTCCCCCC-3'), can serve as reference sequences for future studies using ssDNA-SWCNT sensors to detect 5HT. In this study, we did not find any sequences with a selectivity greater than 1.0, meaning that the fluorescence intensity for 5HT exceeded that for DA.

Comment 7. The article did not conduct any exploratory experiments related to the application of actual samples of this sensor, what do you think are the possible limitations of this sensor in its future application to neurodegenerative diseases?

Answer: Thank you for the suggest and great comment. In our experiments, we did not use actual samples, however we can apply those sensors in brain tissue. Problems with applying the sensor to actual cells include decreased fluorescence intensity, toxicity, biodegradability, and lifetime. We have used this experiment to identify ssDNA sequences that are highly responsive to 5HT, and we expect them to be resistant to changes in intensity. We have revised the manuscript as below.

Changes to manuscript:

Discussion

The biomedical application of SWCNT faces challenges, including concerns about toxicity, the necessity for improved biocompatibility, and ensuring long-term stability. Addressing issues related to biodistribution, immunogenicity, production scalability, and ethical considerations is crucial for advancing the responsible use of SWCNTs in biomedical engineering. Ongoing research efforts are essential to overcome these limitations and unlock the full potential of SWCNTs in clinical applications.

This manuscript is a resubmission of an earlier submission. The following is a list of the peer review reports and author responses from that submission.

Round 1

Reviewer 1 Report

Comments and Suggestions for Authors

This work reports a method to enhance the serotonin-responsive ssDNA-wrapped single-walled carbon nanotube (ssDNA-SWCNT) nanosensors through a combination of directed evolution and machine learning-based prediction. The manuscript writing and figure organization are clear, however there are concerns that needed to be addressed.

-       The authors should discuss the merit of the proposed method with the currently existing approaches for the improvement of sensitivity and selectivity of the sensors.

-       Fluorescent images should be included as a main figure rather than the spectra only.

Comments on the Quality of English Language

The quality of English is moderate, needing to be improved by an editing service.

Author Response

Please see attachment, thank you.

Reviewer 2 Report

Comments and Suggestions for Authors

5HT sensing is not a difficult task for biosensors. This work uses machine learning-based screening and how creativity is reflected. The authors applied directed evolution technology for the optimal ssDNA-SWCNT complex sensor. Need to increase comparison with existing sequences.

P values in Figure 2c and Figure 3d cannot proved the selectivity well. How selective is the sensor under the presence of interfering substances, such as DA, AA, UA?

In addition, the use of SWCNTs in DNA sensors does not have novelty, and I've evaluated the manuscript and have decided it is not the right fit for “Nanomaterials”. My opinion is rejected.

Author Response

Please see attachment, thank you.
